# Active and Happy? Physical Activity and Life Satisfaction among Young Educated Women

**DOI:** 10.3390/ijerph20043145

**Published:** 2023-02-10

**Authors:** Agata Wypych-Ślusarska, Natalia Majer, Karolina Krupa-Kotara, Ewa Niewiadomska

**Affiliations:** 1Department of Epidemiology, Faculty of Health Sciences in Bytom, Medical University of Silesia in Katowice, 41-902 Bytom, Poland; 2Student Scientific Society at the Department of Epidemiology, Faculty of Health Sciences in Bytom, Medical University of Silesia in Katowice, 41-902 Bytom, Poland; 3Department of Biostatistics, Faculty of Health Sciences in Bytom, Medical University of Silesia in Katowice, 41-902 Bytom, Poland

**Keywords:** physical activity, life satisfaction, young adults, SWLS

## Abstract

Background: Physical activity belongs to the group of health-promoting behaviors. It also affects emotional well-being, which is linked to a higher quality of life. Individuals who participate in physical activity practice regardless of age reap several positive health benefits that affect both body and mind. The aim of this study was to assess the life satisfaction of young adults in the context of physical activity undertaken. Material and methods: Study material was collected via anonymous questionnaire surveys among 328 young women (between the ages of 18 and 30 with secondary or higher education) in Poland. Satisfaction with life was assessed using The Satisfaction with Life Scale (SWLS). Statistical calculations were performed using the STATISTICA 13.3 program, Stat Soft Poland. Interdependence of unmeasured characteristics was assessed using the X2 test. Multivariate analysis for the direct effect of physical activity on life satisfaction (LS) and the influence of frequency of physical fitness on life satisfaction was performed based on regular OLS multiple regression. Results: The majority of respondents (74.7%) reported engaging in physical exercise. The mean level of life satisfaction was 4.5 ± 1.1 (on a scale of 1 to 7). Multivariate analysis showed no statistically significant relationship with life satisfaction in the physically active and inactive groups. It was observed that significantly higher levels of life satisfaction were found among respondents: married 5.1 ± 1.1, median = 5.2 (4.5–5.9) compared to single 4.4 ± 1.2, median = 4.6 (3.6–5.2) and in informal relationships 4.5 ± 1.0, median = 4.4 (3.8–5.2); *p* = 0.02; declaring rather good 4.5 ± 1.0, median = 4.6 (3.8–5.2) or very good health 4.8 ± 1.1, median = 5.0 (4.2–5.6) compared to rather poor 4.1 ± 1.0, median = 4.1 (3.4–4.8) and poor health 3.6 ± 1.4, median = 3.1 (2.6–4.4); *p* = 0.0006; rating their physical condition moderately 4.7 ± 1.1, median = 4.8 (4.0–5.6) or highly 4.9 ± 1.0, median = 5.0 (4.3–5.4) compared to rating their fitness low 4.2 ± 0.9, median = 4.2 (3.6–4.8); *p* < 0.0001. Multivariate analyses confirmed a significant effect of marital status and subjective assessment of physical condition on the average level of life satisfaction. Conclusions: Physical activity does not differentiate the level of life satisfaction in the studied group of young women. Marital status and the subjective assessment of physical condition are factors that have a significant impact on the level of satisfaction with the life of young women. Given the beneficial effect of physical activity on the sense of satisfaction with life, which can lead to an increase in its quality, physical activity should be promoted, not only among children but also in the group of young adults.

## 1. Introduction

Physical activity is one of the determinants of quality of life, as well as one of the more effective preventive measures for non-communicable chronic diseases. These conditions include obesity, hypertension, osteoporosis, diabetes, ischemic heart disease, and some cancers, among others. Epidemiological studies have found that the risk of overall mortality can drop by up to 30% with regular physical activity [1,2]. Moderate physical activity reduces the propensity to get sick, as well as influences the course of the disease by alleviating somatic symptoms. The movement also has a positive effect on emotional well-being, which is directly linked to a higher quality of life [3]. Moderate physical activity is associated with physical health, whereas high activity is associated with both health and life satisfaction [4]. Physical inactivity has a huge impact on the risk of chronic non-communicable diseases worldwide, and shortens life expectancy [5]. Scientific evidence further suggests that physical inactivity is responsible for 9% of premature deaths [5]. Analysis of study results from 12 prospective cohorts from the United States and Europe, including a total of 1.44 million participants aged 19 to 98 years, also confirmed the vital importance of physical activity in lowering the risk of many cancer types: esophageal adenocarcinoma (HR = 0.58, CI: 0.37–0.89), liver (HR = 0.73, CI: 0.55–0.98), lung (HR = 0.74, CI: 0.71–0.77), kidney (HR = 0.77, CI: 0.70–0.85), gastric cardia (HR = 0.78, CI: 0.64–0.95), endometrial (HR = 0.79, CI: 0.68–0.92), myeloid leukemia (HR = 0.80, CI: 0.70–0.92), myeloma (HR = 0.83, CI: 0.72–0.95), colon (HR = 0.84, CI: 0.77–0.91), head and neck (HR = 0.85, CI: 0.78–0.93), rectal (HR = 0.87, CI: 0.80–0.95), bladder (HR = 0.87, CI: 0.82–0.92), and breast (HR = 0.90, CI: 0.87–0.93) [6]. 

In addition, physical activity has a positive effect on mental health, cognitive performance, somatic reactions, and performance at work. It has been estimated that as little as 20 min of such activity as swimming, cycling, or running reduces anxiety. In contrast, 25–75 min of moderate-intensity walking affects feelings of agency, mood, and hope, and reduces depressive symptoms and feelings of anxiety [7,8]. Moderate physical activity has a positive effect on the body’s responses to a stressful stimulus. Heart rate and blood pressure heights decrease at rest compared to the pre-exercise resting state [9]. This suggests that relaxation of the heart muscle is occurring. These favorable parameters can last from two to as long as seven hours after a training unit. Highly active individuals have lower somatic responses in response to stressors compared to less physically active individuals. Somatic responses include blood pressure and heart rate, body temperature, and sweat production [9]. In contrast, research on regular physical activity has indicated that active people cope better with stressful situations, and anxiety levels are reduced. This state of affairs may be due to psychological mechanisms. The physiological reactions that occur during physical activity are similar to those of the stress response but without the accompaniment of negative emotions. Consequently, there is an association between somatic symptoms and non-threatening situations, and consequently tolerance to somatic symptoms increases, and the person is better able to manage his or her reactions [9]. Physical activity also enhances cognitive performance, which consists of long-term memory, planning, and daily problem-solving skills. Research has also shown that physically active people are calmer, more productive, and less likely to be exhausted [7,8,9,10]. Feeling calm and relaxed is closely linked to better sleep quality.

Life satisfaction is a subjective ocean of life of a general nature, in which various emotional states appear [11]. Interchangeable names for the above-mentioned phenomenon are happiness, life satisfaction, and mental well-being. Life satisfaction is closely linked to mental health parameters, which include mood, self-esteem, productivity, and feelings of anxiety [12]. 

Meanwhile, it is estimated that one in five people worldwide is physically inactive, and the degree of physical inactivity is increasing with the development of modern technology. A sedentary lifestyle combined with minimal physical activity contributes to a decline in quality of life and is the fourth leading cause of death [13]. A particularly sensitive time to develop negative behaviors associated with physical inactivity is during early adulthood. This may be related to the fact of entering adulthood, starting a career, increasing responsibilities, and reducing leisure time. Due to such developments, when stress levels increase, the frequency of physical activity and life satisfaction levels may decrease (Figure 1). However, it should be noted that this is not a unidirectional relationship. Physical activity and life satisfaction in early life are interrelated and conditioned by many other factors, such as social status, education, economic situation, and personality.

Given these circumstances, a study was planned to assess the life satisfaction of young adults in the context of physical activity undertaken.

The following research questions were posed to address the purpose of the study:Does the amount of time spent on physical activity and its frequency affect feelings of satisfaction with life?Is there a difference in perception of one’s fitness among physically active and inactive people?Does the declared degree of physical fitness affect the subject’s life satisfaction?

## 2. Materials and Methods

### 2.1. Study Area

The survey was conducted from November 2021 to January 2022 using an Internet surveying technique CAWI (Computer Assisted Web Interview). A total of 400 surveys were collected, and 328 surveys were finally qualified for analysis, as 72 did not meet the inclusion criteria (women, aged between 18 and 30, secondary or higher education, childlessness) or the questionnaire was filled out incorrectly. The survey was completed by people residing in Poland.

### 2.2. Characteristics of the Study Group

The study included 328 participants—women, between the ages of 18 and 30 with secondary or higher education. The average age was 23.6 ± 3.2 years. The respondents mainly resided in cities (89.6%), with 20.1% of respondents being rural residents. The study group was dominated by the percentage of those in informal relationships (60.7%).

### 2.3. Eligibility Criteria

Due to the method of data collection used (CAWI, Computer-Assisted Web Interview), non-random sampling (voluntary response sampling) was used [14]. Based on the data of the Central Statistical Office on the state and structure of the population and natural movement in the territorial section (as of 31 December 2019), the minimum sample size corresponding to the group of people aged 18–30 living in Poland was calculated [15]. It was determined that under the conditions of a 95% confidence level, 0.5 fractional size, and 5% maximum error, the minimum sample size was 383 respondents. 

The following inclusion criteria were adopted: women, those with secondary or higher education, those between the ages of 18 and 30, those without children, and those who consented to participate in the study. In contrast, the exclusion criteria for the study were: men, those with primary or vocational education, those under 18 and over 30, those with a child/children, and those who did not agree to participate in the study. 

The childlessness criterion was intended to standardize the structure of the study group in terms of daily activities and lifestyle. On the other hand, the determination of the age of youth is a matter of convention, and researchers point to its different ranges falling between 15 and 30 years [16]. In Poland, the compulsory schooling implemented covers people up to the age of 18, which makes it relatively rare for gainful employment undertaken before this limit to be a significant source of income. The Law on Amendments to the Law on Employment Promotion and Labor Market Institutions proposes an upper age limit for young adults of 30 years [16]. This is in line with the steadily lengthening period of education acquisition. Taking into account the above data, the inclusion criterion for the study was the age of 18–30 years. Only well-educated women (secondary or higher) were included in the study due to the heterogeneity of the research group due to gender and level of education.

Participation in the study was anonymous and entirely voluntary. The study complies with the provisions of the Helsinki Declaration. The project of the study in the light of the Act of December 5, 1996, on the professions of doctor and dentist (*Journal of Laws of 2011, No. 277, item 1634, as amended*), is not a medical experiment and does not require the approval of the Bioethics Committee of the Medical University of Silesia in Katowice.

### 2.4. Research Tool

The study was conducted using an author’s survey questionnaire and the Satisfaction with Life Scale (SWLS) [17,18]. The author’s questionnaire included questions about planned physical activity. Respondents were checked to see if they participate in physical activity, at what frequency, how long one training unit lasts on average, and whether the stated frequency is satisfactory to the subject. The reasons for taking up and not taking up physical activity were also asked, and the respondents’ self-assessment of their physical fitness was examined. 

The Satisfaction with Life Scale (SWLS) was also used in the study. It is a general index of feelings of satisfaction with life. Scale reliability, estimated using the Cronbach alpha index in various analyses, ranges from 0.82 to 0.88 [19,20,21,22,23]. It can be used to survey both sick and healthy individuals. The scale contains five statements. Respondents rated each of these 5 statements separately on a scale ranging from 1 (for “completely disagree”) to 7 (for “completely agree”):

The higher the final score, the higher the sense of satisfaction with life. In the end, after adding up the numbers, one could score from 5 to 35 points. The evaluation analysis was performed based on the average level of satisfaction with life calculated as an average value of the total score for the 5 statements:In most ways, my life is close to my ideal.The conditions of my life are excellent.I am satisfied with my life.So far, I have gotten the important things I want in life.If I could live my life over, I would change almost nothing.

### 2.5. Statistical Analyses

Statistical calculations were performed using the STATISTICA 13.3 program, Stat Soft Poland. Numeric-percentage notation-n (%)—was used to describe non-measurable data. For measurable variables, means and standard deviations (X ± SD) and M medians (quartiles Q1-Q3) were given. The interdependence of non-measurable variables was assessed with the X^2^ test. The significance of differences in means was tested by the Student’s t-test or ANOVA depending on the number of groups. In the absence of a normal distribution (Shapiro-Wilk test), the U Mann-Whitney test or the Kruskal-Wallis test was used. The strength of interdependence of measurable variables was described by Spearman’s R’ coefficient with a significance rating.

Multivariate analysis for a direct effect of physical activity on life satisfaction (*LS*) was performed based on regular OLS multiple regression:

Model 1: LS = b_0_ + b_1_ Physical activity + b_2_ Age + b_3_ Gender + b_4_ Residence + b_5_ Education + b_6_ Professional + b_7_ Marital status + b_8_ Chronic disease + b_9_ Subjective assessment of health + b_10_ Subjective assessment of physical fitness.

The influence of frequency of physical fitness on life satisfaction (*LS*) was analyzed by the OLS model:

Model 2: LS = b_0_ + b_1_ Age + b_2_ Gender +b_3_ Residence + b_4_ Education + b_5_ Professional + b_6_ Marital status + b_7_ Chronic disease + b_8_ Subjective assessment of health + b_9_ Subjective assessment of physical fitness + b_10_ Frequency of practicing physical activity + b_11_ Unit training.

Model 1′ (based on Model 1) and Model 2′ (based on Model 2) present the impact on life satisfaction (*LS*) with the exclusion of confounding factors directly related to physical activity.

Statistical significance was determined at *p* < 0.05. 

## 3. Results

The survey group was dominated by those whose doctor had never diagnosed a chronic disease (80.8%). In contrast, respondents with chronic diseases most often mentioned: metabolic diseases (n = 36, 11.0%), respiratory diseases (n = 10, 3.0%), depression (n = 5, 1.5%), cardiovascular diseases (n = 3, 0.9%), autoimmune diseases (n = 2, 0.6%), and gastrointestinal diseases (n = 1, 0.3%). Respondents indicated a subjective assessment of their health at the following levels: rather bad (n = 8, 2.4%), average (n = 66, 20.1%), rather good (n = 192, 58.5%), and very good (n = 62, 18.9%). According to the answers to the questionnaire questions “do you smoke cigarettes” and “do you drink alcohol.”, cigarette smoking was declared by 94 (28.7%) respondents and alcohol drinking by 292 (89.0%).

The vast majority of respondents (n = 245, 74.7%) reported engaging in physical activity, whereas the minority (n = 83, 25.3%) of respondents did not engage in physical activity practice at all. Engaging in physical activity was unrelated to marital status, smoking, drinking alcohol, and the presence of chronic disease (Table 1). In contrast, living in urban areas, having a higher level of education, and having a job were associated with a higher percentage of those participating in physical activity. Women who practiced physical activity were significantly older than those who did not engage in physical activity. In addition, almost half of the respondents who rated their health poorly or very poorly did not engage in physical activity. 

Among those who participated in physical activity practice (N = 245), the vast majority undertook physical activity several or more times a week (n = 197, 80.4%), (n = 39, 15.9%) once a week, whereas the smallest percentage were those who trained occasionally (n = 9, 3.7%). Workouts of less than 1 h were declared by (n = 20, 8.2%) people, those lasting 1 h (n = 157, 64.1%), 2 h or more (n = 68, 27.7%). 

Participation in physical activity practice was significantly related to subjective assessment of physical condition (*p* < 0.0001), and subjective assessment of well-being (*p* = 0.04) (Figure 2).

The mean level of satisfaction with life as assessed by the Satisfaction with Life Scale (SWLS) questionnaire in the study group was 4.5 ± 1.1 (on a scale of 1 to 7). The frequency of responses to the questionnaire questions is shown in Figure 3. There was a significant correlation between the subjective assessment of satisfaction with life and the average score of the SWLS questionnaire (Spearman’s R’ = 0.61, *p* < 0.0001).

There was a significant difference in the average level of life satisfaction among those who practice physical activity 4.6 ± 1.1, median = 4.6 (3.8–5.4), and among those who undertake physical activities 4.3 ± 1.1, median = 4.2 (3.6–5.0), *p* = 0.02. Among the total respondents, significant differences in the level of life satisfaction were observed among groups by marital status, subjective assessment of health, and subjective assessment of physical condition. It was observed that significantly higher levels of life satisfaction were found among respondents: married 5.1 ± 1.1, median = 5.2 (4.5–5.9) compared to single 4.4 ± 1.2, median = 4.6 (3.6–5.2) and in informal relationships 4.5 ± 1.0, median = 4.4 (3.8–5.2); *p* = 0.02; declaring rather good 4.5 ± 1.0, median = 4.6 (3.8–5.2) or very good health 4.8 ± 1.1, median = 5.0 (4.2–5.6) compared to rather poor 4.1 ± 1.0, median = 4.1 (3.4–4.8) and poor health 3.6 ± 1.4, median = 3.1 (2.6–4.4); *p* = 0.0006; rating their physical condition moderately 4.7 ± 1.1, median = 4.8 (4.0–5.6) or highly 4.9 ± 1.0, median = 5.0 (4.3–5.4) compared to rating their fitness low 4.2 ± 0.9, median = 4.2 (3.6–4.8); *p* < 0.0001.

Multivariate analyses confirmed a significant effect of marital status and subjective assessment of physical condition on the average level of life satisfaction according to the SWLS scale (Table 2). 

Among the factors having a significant impact on the level of life satisfaction, the following were finally selected: marital status and subjective assessment of physical condition (Figure 4).

## 4. Discussion

The main purpose of this study was to assess the life satisfaction of young women in the context of physical activity. Life satisfaction is the product of many different factors and is undoubtedly related to physical activity. However, the most commonly studied group in terms of the impact of the activity on health and overall satisfaction is older adults [19,24]. Thus, it was important to study the impact of the activity on life satisfaction in women who are just entering adulthood and are more likely than older people to enjoy better health and greater energy. Although an increasing number of scientific papers have addressed this topic, there is still a paucity of research in this area among young women [12]. 

The results of the conducted study (multivariate analysis) showed no statistically significant difference between the average level of life satisfaction in the physically active women and the women not undertaking such activity. This observation seems surprising since the results of other studies indicate that physical activity is one of the lifestyle elements that positively affect the level of satisfaction with life [19,24,25]. The Social Diagnosis survey, which was conducted in Poland periodically between 2000 and 2015, showed that physically inactive people were significantly more likely to be dissatisfied with their health, and their level of satisfaction with their current life, as well as the previous year, was lower than that of physically active people [26]. In every year in which measurements were taken, the percentage of those who were dissatisfied and very dissatisfied with their health was higher among the physically inactive [26]. At the same time, it should be noted that the very concept of “life satisfaction” is vague and variously understood even among psychologists. 

Often, life satisfaction is associated with well-being, which can be understood in hedonic terms (a focus on happiness and the pursuit of pleasure) or eudaimonic terms (a focus on meaning and self-actualization) [27]. This division, therefore, suggests that motivation is an important factor influencing feelings of satisfaction with life, including in combination with other determinants, including physical activity [27]. This is also due to the fact that satisfaction and contentment are not constant, but change over the course of life [28]. Physical activity in middle age and old age can improve physical functioning, improve and/or maintain health, and thus enable individuals to achieve plans and goals, which in turn will contribute to life satisfaction [29]. In the case of young women, whose health is usually very good and who do not suffer from chronic illnesses, life satisfaction may not be related to physical exercise, as the achievement of goals, and the realization of plans are independent of this type of activity. The present study focused on a group of young women, hence, most likely, the observation indicates that there are no differences in average life satisfaction between those who are physically active and those who are not. Multivariate analysis confirmed the importance of marital status (married women) and a high or moderate self-assessment of physical fitness for higher levels of life satisfaction. In contrast, it was observed that in the group of exercisers, life satisfaction depended on marital status and self-assessment of physical fitness. This result may suggest a reinforcing effect of physical activity with the cooperation of other factors on life satisfaction. It is possible that the explanation for the above situation is found in the sense of agency and individual motivation. During physical exercise, the sense of proficiency increases, and the skills acquired during this time build a person’s belief that he is capable of conquering the goals he has set for himself. As a result, he has a better perception of himself, is satisfied, has a desire to constantly develop, and is more motivated to overcome challenges in other areas of life as well. However, for this analogy to work, physical activity must have value for the person who undertakes the activity [30]. It seems, therefore, that the motivational factor plays a key role here. The results of a study conducted in a group of physically active elderly people showed that those with a higher level of education have greater motivational deposits, which can also affect the willingness to undertake physical activity, its frequency, and ultimately the level of satisfaction with life [27]. Motives, goals set, and, indirectly, life values can affect life satisfaction. Physical activity is most likely a variable that reinforces the demographic factors analyzed, thus highlighting the differences in the group of people undertaking it. In addition, results from other studies have shown that demographic characteristics are related to feelings of happiness. Higher subjective well-being or happiness is more often experienced by women, those who are married or in committed relationships, and those with higher education [30,31,32,33,34,35,36]. These are factors that influence greater health awareness and lifestyle, and build a sense of emotional and economic security [34].

In the self-report study, higher levels of life satisfaction depended on higher self-assessment of health status and higher self-assessment of physical fitness. However, this observation may be a two-way street, as overall well-being and thus life satisfaction will depend on health status and fitness level. People with lower levels of physical fitness, not only due to the presence of infirmity or illness, may experience lower life satisfaction [37]. Sometimes the decisive factor is the ability to achieve the set goal—in the case of physical activity, when this goal is not yet achieved, as a result of, for example, lower fitness, the level of life satisfaction will be correspondingly lower. This hypothesis seems plausible given the result of a study conducted in a group of 2345 Taiwanese adults: the highly and moderately active groups had significantly higher life satisfaction compared to the low-active group [25]. Research on the effect of physical activity on satisfaction has shown that it has a significant effect on feelings of happiness for up to four years [37]. A squat test of 1 min was associated with feelings of happiness. The test highlighted strength coming from the abdominal muscles and was correlated with physical appearance. For women, in particular, a healthy and attractive appearance increases self-esteem and subjective levels of happiness. The study also detected an association between flexibility levels and satisfaction [38,39]. This may be related to the effect of body flexibility on positive health manifested by a reduction in muscle pain. This is important because a reduction in pain during daily functioning can have a significant impact on feelings of enjoyment of life [38]. To enhance self-satisfaction, reduce negative emotions, and positively affect physical fitness, regularity in activity two to three times a week for about six months is enough. This is influenced by both the reactions occurring in the human body and the building of interpersonal relationships that often accompanies physical activity [40].

## 5. Strengths and Limitations

A strength of the study conducted is the accurate selection of the group eliminating potential confounding factors such as age, gender, and having children. These factors affect lifestyle and daily activity. The homogenization of the group made it possible to point out the essence of the physical activity undertaken by the group of young women. On the other hand, the criteria used resulted in a group that differs in certain characteristics from the rest of the population at that age, and therefore cannot be considered a representative group. Thus, the results and conclusions should be referred only to the surveyed individuals. In addition, the model of the study used (cross-sectional study) does not allow us to examine the impact of physical activity on life satisfaction, whereas it was possible to determine such a relationship. However, it should be borne in mind that not every relationship of statistical association automatically implies a cause-and-effect relationship. However, taking into account other demographic factors, it can be concluded that physical activity is a significant factor in enhancing and differentiating feelings of life satisfaction across subgroups.

In addition, when interpreting the present data, it is important to take into account the timing of the study, which falls during the COVID-19 pandemic period. Research data indicate a link between daily activity, lifestyle, including physical activity, and the austerity resulting from the pandemic [41,42,43,44]. A study conducted in Sweden showed that people reporting reduced physical activity simultaneously reported lower levels of life satisfaction [41]. In the present study, however, we did not analyze the relationship between the pandemic and physical activity levels, nor did the research protocol allow such an analysis. On the one hand, this can be seen as a limitation of the study, but on the other hand, given the results of similar studies considering the pandemic situation, the conclusions drawn seem relevant in the context of mental health. Higher life satisfaction in the physically active group may be a protective factor in the event of a situation of social strictures.

## 6. Conclusions

Physical activity does not differentiate the level of life satisfaction in the studied group of young women; however, differences were observed with regard to marital status in the subgroup of the physically active woman. Higher levels of life satisfaction in the group of exercisers depend on self-assessment of health status and physical fitness. Marital status and the subjective assessment of physical condition are factors that have a significant impact on the level of satisfaction with the life of young women. In view of the beneficial effect of physical activity on the sense of satisfaction with life, which can lead to an increase in its quality, physical activity should be promoted, not only among children but also in the group of young women.

## Figures and Tables

**Figure 1 ijerph-20-03145-f001:**
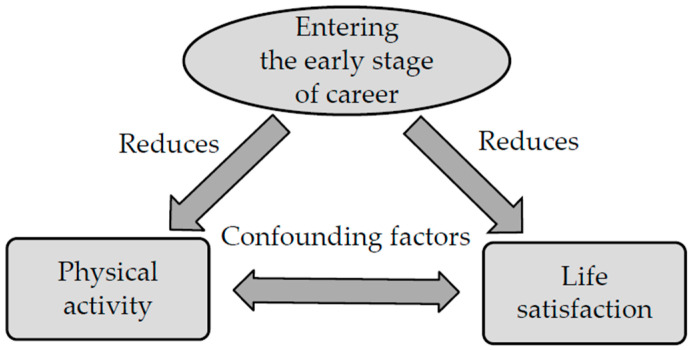
The confounded causality due to the self-selection bias (endogeneity) of “being physically active”.

**Figure 2 ijerph-20-03145-f002:**
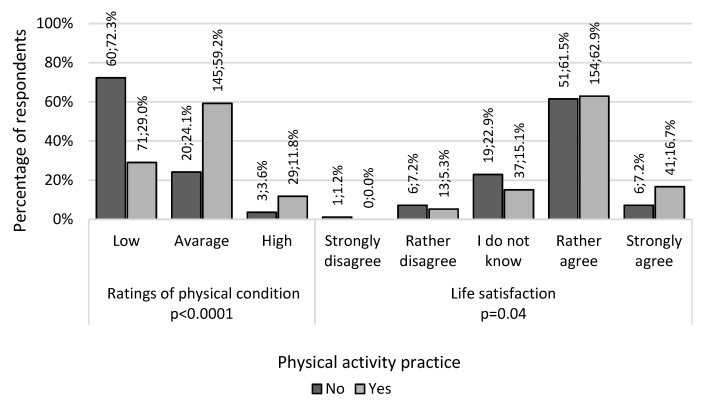
Summary of subjective ratings of physical condition and life satisfaction in relation to physical activity practice participation.

**Figure 3 ijerph-20-03145-f003:**
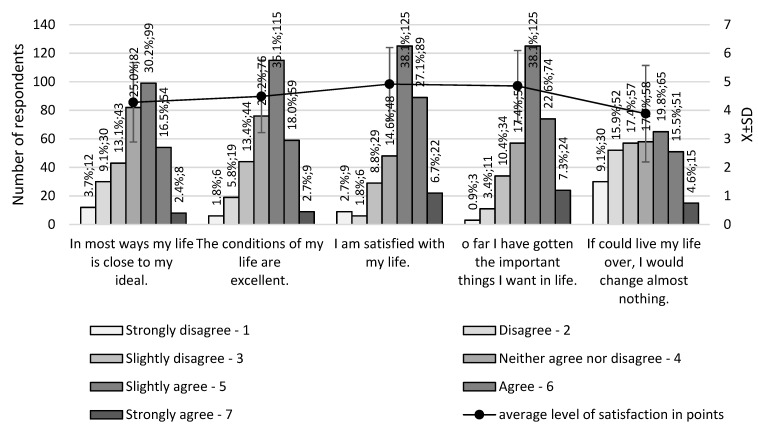
Results of the SWLS questionnaire in a group of study subjects.

**Figure 4 ijerph-20-03145-f004:**
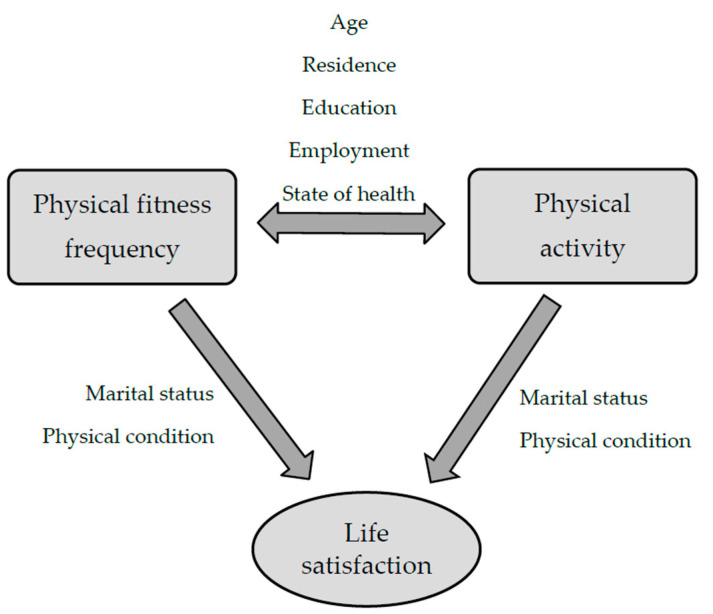
Factors affecting the level of life satisfaction according to the SWLS scale.

**Table 1 ijerph-20-03145-t001:** Characteristics of respondents by physical activity practice.

Variables		Total N = 328	Physical Activity Practice		*p*-Value
			No n (%)	Yes n (%)	
Age	[years].	23 (21–26)	22 (21–25)	24 (21–26)	0.01
Location residence	Village	66 (20.1)	27 (40.9)	39 (59.1)	0.004
	City up to 150,000.	86 (26.2)	17 (19.8)	69 (80.2)	
	A city of more than 150,000.	176 (53.7)	39 (22.2)	137 (77.8)	
Education	Medium	167 (50.9)	55 (32.9)	112 (67.1)	0.001
	Higher	161 (49.1)	28 (17.4)	133 (82.6)	
Employment status	Pupil/student	175 (53.4)	61 (34.9)	114 (65.1)	<0.0001
	Employed/working	153 (46.6)	22 (14.4)	131 (85.6)	
Marital status	Free	109 (33.2)	28 (25.7)	81 (74.3)	0.84
	In an informal relationship	199 (60.7)	51 (25.6)	148 (74.4)	
	In marriage	20 (6.1)	4 (20.0)	16 (80.0)	
Smoking	Not	234 (71.3)	54 (23.1)	180 (76.9)	0.14
	Yes	94 (28.7)	29 (30.9)	65 (69.1)	
Alcohol	Not	36 (11.0)	12 (33.3)	24 (66.7)	0.24
	Yes	292 (89.0)	71 (24.3)	221 (75.7)	
Disease chronic	Not	265 (80.8)	64 (24.2)	201 (75.8)	0.32
	Yes	63 (19.2)	19 (30.2)	44 (69.8)	
Subjective assessment state of health	Rather badly	8 (2.4)	4 (50)	4 (50)	<0.0001
	On average	66 (20.1)	30 (45.5)	36 (54.6)	
	Rather well	192 (58.5)	41 (21.4)	151 (78.7)	
	Very good	62 (18.9)	8 (12.9)	54 (87.1)	

Data presented as numbers and percentages in rows-n (%).

**Table 2 ijerph-20-03145-t002:** The results of multivariate analysis for the level of satisfaction according to the SWLS scale in relation to physical activity practice or frequency of physical fitness.

The Average Level of Satisfaction According to SWLS Scale		Model 1		Model 1′		Model 2		Model 2′	
Data		*b*	*p*-Value	*b*	*p*-Value	*b*	*p*-Value	*b*	*p*-Value
Physical activity practice	Not	-	-			-	-	-	-
	Yes	0.01	0.87	0.03	0.67				
Age	[years]	0.04	0.19	-	-	0.18	0.31	-	-
Location residence	Village	-	-	-	-	-	-	-	-
	City up to 150,000	0.08	0.37			0.15	0.78		
	City of more than 150,000	−0.05	0.52			−0.04	0.94		
Education	Medium	-	-	-	-	-	-	-	-
	Higher	0.03	0.68			0.18	0.69		
Activity professional	Pupil/student	-	-	-	-	-	-	-	-
	Employed/working	−0.15	0.06			−0.69	0.17		
Marital status	Free/non	-	-	-	-	-	-	-	-
	In a relationship Informal	−0.12	0.23	−0.17	0.09	−0.85	0.16	−0.21	0.06
	In a relationship Married	0.35	0.04	0.41	0.01	1.91	0.06	0.44	0.01
Smoking	Not	-	-	-	-	-	-	-	-
	Yes	−0.03	0.60	−0.04	0,50	−0.10	0.78	−0.04	0.62
Alcohol	Not	-	-	-	-	-	-	-	-
	Yes	0.10	0.30	0.11	0.23	0.36	0.55	0.04	0.76
Disease chronic	Not	-	-	-	-	-	-	-	-
	Yes	0.04	0.62	0.02	0.77	0.31	0.50	0.03	0,76
A subjective assessment of state of health	Rather bad	-	-	-	-	-	-	-	-
	Average	−0.01	0.91			−1.28	0.21		
	Rather good	0.24	0.04			0.54	0.51		
	Very good	0.38	0.01			1.52	0.10		
Subjective assessment ofphysical condition	Low	-	-	-	-	-	-	-	-
	Average	0.03	0.71	0.04	0.62	0.33	0.51	0.07	0.48
	High	0.29	0.03	0.33	0.01	1.32	0.07	0.30	0.04
Frequency of practicing physical activity	Occasionally	-	-	-	-	-	-	-	-
	Once a week					0.27	0.75	0.03	0.86
	Several or more times a week					0.46	0.53	0.16	0.25
Unit training	Less than 1 h	-	-	-	-	-	-	-	-
	1 h					0.01	0.98	0.03	0.80
	2 h and more					0.02	0.98	0.02	0.90

## Data Availability

Not applicable.

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
