# Peer review of "Active and Happy? Physical Activity and Life Satisfaction among Young Educated Women"

_ijerph, 2023, doi:10.3390/ijerph20043145_

Round 1

Reviewer 1 Report

Please see the attached PDF file for my major and minor comments.

Author Response

Dear Editor,

Thank you for your review and all your valuable comments. They are extremely relevant and pertinent. We have included all of them in the revised manuscript and marked the changes in red. The green color indicates the updated calculations.

Major comments:

Ref. 1. We have added two literature items relating to the adult group and the general population [footnotes 5 and 6]. We supplemented the introduction with the following passage:

Page 2, lines 58-70:

Physical inactivity has a huge impact on the risk of chronic non-communicable diseases worldwide, and shortens life expectancy [5]. Scientific evidence further suggests that physical inactivity is responsible for 9% of premature deaths [5]. Analysis of study results from 12 prospective cohorts from the United States and Europe, including a total of 1.44 million participants aged 19 to 98 years, also confirmed the significant importance of physical activity in lower risk of many cancer types: esophageal adenocarcinoma (HR=0.58,CI:0.37-0.89), liver (HR=0.73,CI:0.55-0.98), lung (HR=0.74,CI:0. 71-0.77), kidney (HR=0.77,CI:0.70-0.85), gastric cardia (HR=0.78,CI:0.64-0.95), endometrial (HR=0.79,CI:0.68-0.92), myeloid leukemia (HR=0.80,CI:0.70-0.92), myeloma (HR=0.83,CI:0. 72-0.95), colon (HR=0.84,CI:0.77-0.91), head and neck (HR=0.85,CI:0.78-0.93), rectal (HR=0.87,CI:0.80-0.95), bladder (HR=0.87,CI:0.82-0.92), and breast (HR=0.90,CI:0.87-0.93) [6].

Ad.2. These are valid and pertinent observations. In the original excerpt, we put it too one-way, so we added the appropriate addition

(a) Page 3, lines: 104-109

Due to such developments, when stress levels increase, the frequency of physical activity and life satisfaction level may decrease (Figure 1). However, it should be noted that this is not a unidirectional relationship. Physical activity and life satisfaction in early life are interrelated and conditioned by many other factors, such as social status, education, economic situation, and personality.

Added Figure 1 - Page 3, lines 110-112

(b) We removed the following sentence from the introduction:

Therefore, for young adults, taking care of their emotional well-being is an important aspect.

Given the passage completed earlier (from a), this sentence no longer appears to be necessary.

Ad.3. The reviewer's suggestion for in-depth analyses was taken into account.

Suggested multivariate analyses were performed:

Page 5, lines 200-21: Multivariate analysis for direct effect of physical activity on life satisfaction (LS) was performed based on regular OLS multiple regression:

Model 1: LS = b0 + b1 Physical activity + b2 Age + b3 Gender +b4 Residence + b5 Education + b6 Professional + b7 Marital status + b8 Chronic disease + b9 Subjective assessment of health + b10 Subjective assessment of physical fitness.

The influence of frequency of physical fitness on life satisfaction (LS) was analysed by OLS model:

Model 2: LS = b0 + b1 Age + b2 Gender +b3 Residence + b4 Education + b5 Professional + b6 Marital status + b7 Chronic disease + b8 Subjective assessment of health + b9 Subjective assessment of physical fitness + b10 Frequency of practicing physical activity + b11 Unit training

Model 1' (based on Model 1) and Model 2' (based on Model 2) present the impact on life satisfaction (LS) with the exclusion of confounding factors directly related to physical activity.

The results of the calculations are shown in Table 3 .

A summary of the analyses is shown in Figure 4

Ad.4. As suggested, the research group was limited to women with secondary or higher education.

Information regarding the research group was included in the title of the paper:Active and happy? Physical activity and life satisfaction among young educated women

The group selection in the Materials and methods section is included:

The survey was conducted from November 2021 to January 2022 using an Internet surveying technique CAWI (Computer Assisted Web Interview). 400 surveys were collected, and 328 surveys were finally qualified for analysis, as  72 did not meet the inclusion criteria (women, age between 18-30, secondary or higher education, childlessness) or the questionnaire was filled out incorrectly. The survey was completed by people residing in Poland.

The study included 328 participants – women, between the ages of 18 and 30 with secondary or higher education. The average age was 23.6±3.2 years. The respondents mainly resided in cities (89.6%), with 20.1% of respondents being rural residents. The study group was dominated by the percentage of those in informal relationships (60.7%) (Table 1).

The following inclusion criteria were adopted: women, those with secondary or higher education, those between the ages of 18 and 30, those without children, and those who consented to participate in the study. In contrast, the exclusion criteria for the study were: men, those with primary or vocational education, those under 18 and over 30, those with a child/children, and those who did not agree to participate in the study.

Only well-educated women (secondary or higher) were included in the study due to the heterogeneity of the research group due to gender and level of education.

Recalculations were performed based on the reduced study group - the updated results are marked in red.

Ad. 5. We have added a list of SWSL questionnaire questions and information on its reliability.

Scale reliabillity, oszacowana przy pomocy the Cronbach alpha index w róznych analizach wynosi od 0.82 do 0.88 [Hultell, D., & Gustavsson, J. P. (2008). A psychometric evaluation of the Satisfaction with Life Scale in a Swedish nationwide sample of university students. Personality and Individual Differences, 44(5), 1070-1079. DOI:10.1016/j.paid.2007.10.030; Vazquez, C., Duque, A., & Hervas, G. (2013). Satisfaction with life scale in a representative sample of Spanish adults: validation and normative data. Spanish Journal of Psychology, 16(82), 1-15.; Swami, V., & Chamorro-Premuzic, T. (2009). Psychometric evaluation of the Malay Satisfaction with Life Scale. Social Indicators Research, 92(1), 25-33.; Vera-Villarroel, P., Urzúa, A., Celis-Atenas, P. P. K., & Silva, J. (2012). Evaluation of subjective well-being: Analysis of the satisfaction with life scale in Chilean population. Universitas Psychologica, 11(3), 719-727]

  1. In most ways, my life is close to my ideal.
  2. The conditions of my life are excellent.
  3. I am satisfied with my life.
  4. So far, I have gotten the important things I want in life.
  5. If I could live my life over, I would change almost nothing.

Ad.6

Table 1 has been removed and the results moved to the current Table 1 (formerly Table 2).

Ad. 7

We have removed information from the manuscript regarding motives for engaging or not engaging in physical activity.

Ad. 8

I have changed  subjective assessment of fitness in Subjective assessment of physical condition

Ad 9.

Abstract and discussion have been updated

Minor comments:

  1. “Place of study” Rather than presenting a map of the entire Europe, zoom into show how samples distribute within the map of Poland. This makes the graphical presentation more informative.

The survey was anonymous, so no data was collected on the exact place of residence. This information was not relevant to the topic of the survey. Table 1, including the map, has been removed, and the results have been moved to the current Table 1 (formerly Table 2).

  1. “Professional activity” Change it to “Employment status”, which is a more commonly used term in the English world.

I have changed „professional activity” in  „employment status”

  1. “Sex” Do you mean “Biological sex at birth”? If so, make it clear. Also, double-check the consistency of wording throughout this manuscript. Do not use “sex”, “gender”, and “biological sex at birth” interchangeably.

We standardized the nomenclature by using the word „gender”.

  1. P4, L122 “non random sampling was used” Why was a random sampling not conducted? Which “nonrandom” sampling strategy did you follow? Citations are needed to justify the choice of such a non-random sampling.

We opted for non-random sampling, due to the fact that in the case of our study, a random sample based on probability is not feasible due to the location of the survey (online survey). We used voluntary response sampling. We have supplemented this information in the manuscript

Due to the method of data collection used (CAWI, Computer-Assisted Web Interview), non random sampling (voluntary response sampling) was used [Saunders, M., Lewis, P. & Thornhill, A. (2012) “Research Methods for Business Students” 6th edition, Pearson Education Limited.]

  1. We have standardized the vocabulary throughout the text in accordance with the Reviewer's suggestion, corrected the figure numbering, and translated the figure data into English.

Reviewer 2 Report

Dear Authors,

Thank you for letting me review this paper.

I think there is a big problem to measure Physical activity and life satisfaction among young adults during 2021 and 2022, not to mention COVID-19 which affected all physical activities and life satisfaction around the world during this time. Therefore it is difficult to review this paper. Another big issue is the skew gender distribution, which is not mentioned in the limitations.<

For instance, are there articles about Physical activity and life satisfaction during this time;

Eek, F.; Larsson, C.; Wisén, A.; Ekvall Hansson, E. Self-perceived changes in physical activity and the relation to life satisfaction and rated physical capacity in Swedish adults during the COVID-19 pandemic—a cross-sectional study. Int. J. Environ. Res. Public Health 2021, 18, 671.

Blom, V.; Lönn, A.; Ekblom, B.; Kallings., L.V.; Väisanen, D.; Hemmingsson, E.; Andersson, G.; Wallin, P.; Stenling, A.; Ekblom, Ö.; et al. Lifestyle habits and mental health in light of the two COVID-19 pandemic waves in Sweden, 2020, Int. J. Environ. Res. Public Health 2021, 18, 3313

Dahlen, M.; Thorbjørnsen, H.; Sjåstad, H.; von Heideken Wågert, P.; Hellström, C.; Kerstis, B.; Lindberg, D.; Stier, J.; Elvén, M. Changes in physical activity are associated with corresponding changes in psychological well-being: A pandemic case study. Int. J. Environ. Res. Public Health 2021, 18, 10680.

Elvén, M.; Kerstis, B.; Stier, J.; Hellström, C.; von Heideken Wågert, P.; Dahlen, M.; Lindberg, D. Changes in Physical Activity and Sedentary Behavior before and during the COVID-19 Pandemic: A Swedish Population Study. Int. J. Environ. Res. Public Health 2022, 19(5).

·                     Page 1 line 25: why p value determined in the abstract?

·                     Overall please include p < .003.

·                     Page 1 line 43: Why not everybody?

·                     Page 2 line 60: Is this all from the same reference?
Moderate physical activity has a positive effect on the body's responses to a stressful stimulus. Heart rate and blood pressure heights decrease at rest compared to pre-exercise resting state. This suggests that the relaxation of the heart muscle is occurring. These favorable parameters can last from two to as long as seven hours after a training unit. Highly active individuals have lower somatic responses in response to stressors compared to less physically active individuals. Somatic responses include blood pressure and heart rate, body temperature, and sweat production. In contrast, research on regular physical activity has indicated that active people cope better with stressful situations, and anxiety levels are reduced. This state of affairs may be due to psychological mechanisms. The physiological reactions that occur during physical activity are similar to those of the stress response but without the accompaniment of negative emotions. Consequently, there is an association between somatic symptoms and non-threatening situations, and consequently tolerance to somatic symptoms increases, and the person is better able to manage his or her reactions [7].

·                     Why not use PA for physical activity?

·                     Page 5 line 187: Definition of alcohol drinking and smoking.

·                     Page 7 line 225: Please write in English.

·                     Page 8. Difficult to read your tables, maybe every second row can be greyed.

·                     Page 10. Why use he?

Author Response

Dear Reviewer,

Thank you very much for reading our manuscript AND for your valuable comments. We refer to the most important ones below:

I think there is a big problem to measure Physical activity and life satisfaction among young adults during 2021 and 2022, not to mention COVID-19 which affected all physical activities and life satisfaction around the world during this time. Therefore it is difficult to review this paper. Another big issue is the skew gender distribution, which is not mentioned in the limitations.<

We have modified our manuscript significantly. We limited the study group to women with a high school education or higher, which undoubtedly allows us to remove variables that distort the results of the study.

We included the period of the study (pandemic) in the section: limitations and strengths. We have cited the following publications:

Eek, F.; Larsson, C.; Wisén, A.; Ekvall Hansson, E. Self-perceived changes in physical activity and the relation to life satisfaction and rated physical capacity in Swedish adults during the COVID-19 pandemic—a cross-sectional study. Int. J. Environ. Res. Public Health 202118, 671.

Blom, V.; Lönn, A.; Ekblom, B.; Kallings., L.V.; Väisanen, D.; Hemmingsson, E.; Andersson, G.; Wallin, P.; Stenling, A.; Ekblom, Ö.; et al. Lifestyle habits and mental health in light of the two COVID-19 pandemic waves in Sweden, 2020, Int. J. Environ. Res. Public Health 202118, 3313

Dahlen, M.; Thorbjørnsen, H.; Sjåstad, H.; von Heideken Wågert, P.; Hellström, C.; Kerstis, B.; Lindberg, D.; Stier, J.; Elvén, M. Changes in physical activity are associated with corresponding changes in psychological well-being: A pandemic case study. Int. J. Environ. Res. Public Health 202118, 10680.

Elvén, M.; Kerstis, B.; Stier, J.; Hellström, C.; von Heideken Wågert, P.; Dahlen, M.; Lindberg, D. Changes in Physical Activity and Sedentary Behavior before and during the COVID-19 Pandemic: A Swedish Population Study. Int. J. Environ. Res. Public Health 2022, 19(5).

  • Page 1 line 25:why p value determined in the abstract?

P value was removed from the abstract.

                     Page 1 line 43: Why not everybody?

In the study, we focused on young women, and we relate our conclusions to this group. Hence the conclusion that physical activity should also be particularly promoted among young women. As we mentioned in our manuscript, these individuals are entering the labor market, a time of significant changes in their lifestyle. This is associated with the abandonment of physical activity or not engaging in it at all. Of course, we agree that physical activity is important at any age, and this is an indisputable truth. However, because of the analysis of young women, we also relate our findings to this group.

                     Page 2 line 60: Is this all from the same reference?
Moderate physical activity has a positive effect on the body's responses to a stressful stimulus. Heart rate and blood pressure heights decrease at rest compared to pre-exercise resting state. This suggests that the relaxation of the heart muscle is occurring. These favorable parameters can last from two to as long as seven hours after a training unit. Highly active individuals have lower somatic responses in response to stressors compared to less physically active individuals. Somatic responses include blood pressure and heart rate, body temperature, and sweat production. In contrast, research on regular physical activity has indicated that active people cope better with stressful situations, and anxiety levels are reduced. This state of affairs may be due to psychological mechanisms. The physiological reactions that occur during physical activity are similar to those of the stress response but without the accompaniment of negative emotions. Consequently, there is an association between somatic symptoms and non-threatening situations, and consequently tolerance to somatic symptoms increases, and the person is better able to manage his or her reactions [7].

Yes, this excerpt refers to one citation [7]. However, for the sake of clarity, we have supplemented the earlier passages of the aforementioned text with a bibliographic footnote.

  • Why not use PA for physical activity?

Physical activity, according to the rules of grammar, does not require the use of an acronym, so we did not choose to replace the phrase with PA

  • Page 5 line 187: Definition of alcohol drinking and smoking.

Cigarette smoking and alcohol consumption were defined based on positive responses to questionnaire questions: Do you smoke cigarettes (Yes/No), Do you drink alcohol (Yes/No). We did not ask about the frequency of alcohol drinking and the number of cigarettes smoked, nor about past smoking. We made changes in the manuscript:

According to the answers to the questionnaire questions "do you smoke cigarettes" and "do you drink alcohol," cigarette smoking was declared by 113 (29.4%) respondents and alcohol drinking by 332 (86.2%).·                 

    Page 7 line 225: Please write in English.

The information in the figure has been translated into English

  • Page 8. Difficult to read your tables, maybe every second row can be greyed.

Re-analyses were conducted for a group of young women, with two multivariate analysis models.

Greetings, Authors 

Round 2

Reviewer 1 Report

This second draft responded to all my comments successfully. And the presentation of results got improved significantly. The only issue is those red percentages in Figures 2 and 3. Please move all percent labels of bars to the top of the bars, given the journal is black-and-white printed. Any number inside a bar will be tough to see on a mono-color page. 

Thanks so much for your efforts in replying to my comments.

Author Response

Dear Reviewer,
Thank you very much for reviewing our manuscript again. We have taken into account your valuable comments, with which we have improved the quality of our study. We are grateful for its recognition.
Greetings